# Characterization of Ionic Transport in Li_2_O-(Mn:Fe)_2_O_3_-P_2_O_5_ Glasses for Li Batteries

**DOI:** 10.3390/ma15228176

**Published:** 2022-11-17

**Authors:** Young-Hoon Rim, Chang-Gyu Baek, Yong-Suk Yang

**Affiliations:** 1College of Liberal Arts, Semyung University, Jecheon 27136, Republic of Korea; 2Department of Nanoenergy Engineering, College of Nanoscience and Nanotechnology, Pusan National University, Busan 46241, Republic of Korea

**Keywords:** Li_2_O-(Mn:Fe)_2_O_3_-P_2_O_5_ glass, conductivity, Cole–Cole plot, power–law, activation energy

## Abstract

We present a systematic study of the lithium-ion transport upon the mixed manganese-iron oxide phosphate glasses 3Li_2_O-xMn_2_O_3_-(2-x)Fe_2_O_3_-3P_2_O_5_(LM_x_F_2−x_PO; 0≤ x ≤2.0) proposed for the use in a cathode for lithium secondary batteries. The glasses have been fabricated using a solid reaction process. The electrical characteristics of the glass samples have been characterized by electrical impedance in the frequency range from 100 Hz to 30 MHz and temperature from 30 °C to 240 °C. Differential thermal analysis and X-ray diffraction were used to determine the thermal and structural properties. It has been observed that the dc conductivity decreases, but the activation energies of dc and ac and the glass-forming ability increase with the increasing Mn_2_O_3_ content in LM_x_F_2−x_PO glasses. The process of the ionic conduction and the relaxation in LM_x_F_2−x_PO glasses are determined by using power–law, Cole–Cole, and modulus methods. The Li^+^ ions migrate via the conduction pathway of the non-bridging oxygen formed by the depolymerization of the mixed iron–manganese–phosphate network structure. The mixed iron–manganese content in the LM_x_F_2−x_PO glasses constructs the sites with different depths of the potential well, leading to low ionic conductivity.

## 1. Introduction

The early research by Goodenough et al. on the phospho-olivines as cathodes has extended LiFe_1−x_Mn_x_PO_4_ to be a positive electrode for lithium secondary batteries [1,2,3]. Since LiMPO_4_ compounds (M = Fe, Mn, Co, or Ni) are appearing features of high lithium bulk mobility, advantageous of safety, thermal and chemical stability, environmental property, and cost effectiveness [4].

Wang et al. reported that orthorhombic LiFePO_4_ has a space group of *Pnma* [5]. Using X-ray absorption spectroscopy and ab initio density functional simulations. Shirakawa et al. investigated the electrochemical reaction of Li insertion on monoclinic with the space group *P2_1_/n* Li_3_Fe_2_(PO_4_)_3_, and it was confirmed that most of Fe^3+^ were reduced to Fe^2+^ with the variation of their electronic *d* orbital [6]. Carbon-coated LiFePO_4_ has been successfully commercialized because it has an electric conductivity of ~10^−1^ S/cm at 30 °C, and LiMnPO_4_ also has brought great attention due to its higher energy density connected to Mn^2+^/Mn^3+^ redox voltage of 4.2 V (LiFePO_4_, Fe^2+^/Fe^3+^; 3.5 V) vs. Li/Li^+^ [7,8,9].

It is emphasized that the key challenges for cathode are mainly valuable and significantly control the cycle life and energy density. To maintain long-term cycling stability, Zuo et al. prepared a nanoporous LiMn_0.8_Fe_0.2_PO_4_ by a facile ion-exchange solvothermal method [10]. Lai and co-authors presented that the crystal structure of manganese iron mixed oxide nanoparticles changed from a single phase of γ-Fe_2_O_3_ to bixbyite of Fe_2_O_3_ and Mn_2_O_3_ with increasing the Mn content [11].

Significant progress in the electrochemical performance has been achieved in the research of lithium secondary batteries such as reducing particle size, surface coating with electronic-conducting species, and cation doping [9,10]. Particle size reduction with the increase in the effective surface area shortens the diffusion length for lithium-ion migration.

In addition, the emergence of nanocrystals within a glass matrix exhibits the superior properties of materials, such as high thermal and chemical stability, high mechanical strength, and electrochemical performance. Designing nanoparticles for practical applications requires knowledge and control of how their desired properties related to their composition and structure. The advantages of using glass in synthesizing crystalline composites are based on such characteristics as the selection of various atomic species, control of fabricating temperature by selecting adequate glass former, and easy variation of producing the nanocrystalline constituents with isotropically distributed in the network structure [12,13,14]. A mixture of such materials would be expected to have a broad range of surface chemical properties.

Rim et al. reported in a previous paper that the addition of lithium oxide (e.g., Li_2_O) in xLi_2_O-2Fe_2_O_3_-3P_2_O_5_ glasses increased the depolymerization of the iron phosphate in the glass structure with the formation of non-bridging oxygen (NBO) by the conversion of Q^2^ to Q^1^ and Q^0^ units [15]. The notation Q^n^ denotes the structural unit of the glass network and n is the number of bridging oxygens per structure unit. The migration of ions in the glass network occurs by the thermally-driven hops between NBOs sites via the lowest potential channels [16]. In view of the energy landscape, the potential well depth depends on the Coulomb interaction between ions in the glass [17]. In the case of low concentrations of ions, electric conduction is mostly controlled by the long-range Coulomb interaction between migrating ions. However, for higher concentrations of ions, the network modification gives rise to a significant effect on the conducting pathways [17]. However, Rim et al. reported that the low electrical conduction in the lithium silicate glasses is originated from the partial transport of Li^+^ ions through the NBO sites [18]. Therefore, it is a great challenge to investigate the process of fast ion transport in mixed iron–manganese oxide glasses, because the ionicity of manganese and iron is acting differently to the surrounding phosphate network, resulting in the structural modification of the glass network. At present there are some works on material characteristics of mixed iron–manganese oxide nanoparticles, relying on the crystalline state [10,11]. Glass has many advantages in synthesizing a special crystalline composite material. However, the nature on the correlation between phosphate glass structure and ionic transport properties in the lithium-iron–manganese–phosphate glasses is unknown. In this connection, we have verified the influence of manganese additives on the electrical properties of the Li-ion transport upon the mixed manganese-iron oxide in phosphate glass by the estimation of charge carrier density.

In this work. we prepare lithium-iron–manganese–phosphate glasses of 3Li_2_O-xMn_2_O_3_-(2-x)Fe_2_O_3_-3P_2_O_5_ (LM_x_F_2−x_PO), where 0≤ x ≤2.0 and measured the electrical impedance and thermal properties. It is observed that the dc conductivity decreases but the activation energies of dc and ac, and the glass-forming ability, increase with increasing content of the Mn_2_O_3_ in LM_x_F_2−x_PO glasses. The process of the ionic conduction and the relaxation in LM_x_F_2−x_PO glasses are described by using Cole–Cole, power–law, and modulus methods. We have calculated the number density of ionic carriers in the LM_x_F_2−x_PO glasses using the modified fractional Rayleigh equation. Here we have correlated ion migrations with the glassy network through the modification of the local structure. Our present work for ionic transport in the mixed iron–manganese oxide glasses may suggest useful information on the complicated mechanism of ion dynamics for practical applications and the design of improved characteristics in solid electrodes.

## 2. Experimental and Theoretical Background

### 2.1. Experimental Procedure

High grades of Li_2_CO_3_, Fe_2_O_3_, Mn_2_O_3_, and NH_4_H_2_PO_4_ were used as starting materials to prepare glass samples. The glasses with the molar composition of 3Li_2_O-xMn_2_O_3_-(2-x)Fe_2_O_3_-3P_2_O_5_ (LiM_x_F_2−x_PO; x = 0, 0.25, 0.5, 0.75, 1.0, 1.6, 2.0) were prepared by melting the powders in the air in platinum crucibles with heat treating at 1200 °C in the electric furnace. The melts were quenched between brass plates and pressed to have the desired shape and thickness. Then, the quenched samples were heat treated at 300 °C for 1 h in the furnace.

The glass samples were ~0.6 ± 0.1 mm in thickness with a black or dark brown color. The glass of LM_x_F_2−x_PO (0≤ x ≤1.0) was not transparent but lustrous with a black. Meanwhile, the glass of LM_x_F_2−x_PO (x = 1.6 and 2.0) was transparent with a dark brown.

Glass powders of the crunched pellet samples were used for differential thermal analysis (DTA). To get the thermal properties, we carried out the differential thermal analysis (DTA, TG-DTA 2000s; Mac Sci.) measurement with the heating rate of 10 °C/min from 30 °C to 1050 °C and used an empty alumina crucible as a reference. The typical temperatures occurring on transition, the glass transition T_g_, the crystallization *T*_c_, the melting *T_m_*, and the maximum of the peak T_x_, were determined with an extrapolation method.

As-quenched glasses were checked with DTA and X-ray diffraction (XRD; Miniflex II, Rigaku) with CuKα radiation (λ = 1.5406 Å). The XRD patterns were obtained in the range of 10~80° with a 0.05° step and the 3 s counting time. Measurement was repeated 5 times to enhance the signal to noise ratio.

Glass samples were attached with a diameter of 3.0 mm of circular platinum electrodes and a lead with silver paste, and the electrical measurement was performed in a furnace from 30 °C to 240 °C with a 10 °C/min heating. The measurement of the electrical impedance was carried out by using the impedance analyzer (4294A, Keysight) in the frequency range of 100~30 MHz.

### 2.2. Theoretical Background

The dielectric constant behavior in disordered materials has been investigated by dielectric spectroscopy. The complex permittivity,${\;\varepsilon }^{*}\left(\omega \right)=\;\varepsilon^{\prime }\left(\omega \right)-i\varepsilon^{\prime \prime }\left(\omega \right)$, can characterize the dipolar relaxation in liquids and solids. The reorientation of permanent dipoles and the electrical conductivity of ion dynamics in the sample show the frequency-dependent features of the sample. We define the electrical impedance as $Z^{*}\left(\omega \right)=\;Z^{\prime }-{\mathrm{iZ}}^{\prime \prime }=U^{*}/I^{*}$, where $U^{*}$ and $I^{*}$ are the voltage and current, respectively. Then the complex impedance can be expressed as the complex permittivity $Z^{*}\left(\omega \right)=1/\left({i\omega \varepsilon }^{*}\right)$, the conductivity $\sigma^{*}\left(\omega \right)=\;\sigma^{\prime }\left(\omega \right)+{i\sigma }^{\prime \prime }\left(\omega \right)=1/Z^{*}\left(\omega \right)$, and the modulus $M^{*}\left(\omega \right)=1/\varepsilon^{*}\left(\omega \right)={\;i\omega Z}^{*}\left(\omega \right)$. Therefore, these expressions of $Z^{*}\left(\omega \right)$ $\sigma^{*}\left(\omega \right)$ and $M^{*}\left(\omega \right)$ are in principle derived from the same experimental data [19,20,21,22].

The Cole–Cole complex impedance is expressed by [23]:(1)$$Z^{*}=\frac{\Delta R}{1+{\left(i\omega \tau \right)}^{\delta }}\times \frac{A}{L}\;,$$
where L, A, τ and ω are the sample thickness, the electrode area, the relaxation time, and the applied field frequency, respectively, and ΔR  is defined as a difference between the resistances at zero and infinity of frequency. A parameter δ $\left(0\leq \delta \leq 1\right)$ is the weight of the distribution of relaxation time. In the Debye system, the complex impedance plot displays the perfect semicircle with the value of δ = 1. When the structure of the system increases the disorder, then the δ decreases gradually. Thus, the value of δ indicates the distribution of potential barriers by the structural network deformation [16].

Based on the modified fractional Rayleigh equation, Rim et al. showed that the universal power–law was derived elegantly. The advantage of the equation is that the expression involves the information of the number density of the ions in the glass system [24]. The driven equation is expressed as follows:(2)Reσω=σdc+Kωs+Aω=σdc1+ωωhs+Aω,
where $K\sim \frac{N_{q}}{\sqrt{2\pi }}\sum_{n=1}^{\infty }\frac{{b\eta }_{\alpha }\alpha^{n-1}\Gamma \left[1+\left(n-1\right)\alpha \right]}{\Gamma \left(n\right)}\sin\left(\frac{n\alpha }{2}\pi \right)$ and the coefficient A, N, q and${\;\eta }_{\alpha }$ denote a constant, the number, the charge, and the frictional constant of ionic transport, respectively. The value of α is defined as $\;\alpha =\frac{1}{D_{f}}-\frac{1}{3}$ in a pathway, the symbol of D_f_ and $\Gamma \left(n\right)$ denotes the fractional dimension and the gamma function, respectively. Here, $\sigma_{\mathrm{dc}}$ is the dc-conductivity and the exponent s is a number within the range of 0 < s=1−α < 1. The hopping frequency,${\;\omega }_{h}$, is defined by Reσωh=2σdc. The value of $\sigma_{\mathrm{dc}}$ in Equation (2) is a value of Reσω in the limit of low frequencies. The term,${\;K\omega }^{s}$, in Equation (2) is related to the ac conductivity of ionic transport at the onset frequency $\omega_{h}$. The constant loss term in Equation (2) is written as Aω [25].

The hopping frequency ω_h_ in Equation (2) follows the Arrhenius relation:(3)$$\omega_{h}=\omega_{0}\exp\left(-\frac{E_{\mathrm{ac}}^{J}}{k_{B}T}\right),$$
where $\omega_{0}$ is a constant and $E_{\mathrm{ac}}^{J}$ is the ac-activation energy in Equation (2).

The hopping frequency, $\omega_{h}$, is a key parameter to determine the number of mobile charge carriers, $\rho_{m}$, in the glass network [15,24]. The relation can be expressed as
(4)$${\ln\omega }_{h}={\ln\omega }_{0}-\frac{E_{\mathrm{ac}}^{J}}{k_{B}T}=\frac{1}{3\left(1-s\right)}\ln\left(\rho_{m}\sqrt{M}\right),$$
where *M* is the atomic mass per mol.

The relationship between complex modulus, M*(ω), and the complex permittivity, $\varepsilon^{*}\left(\omega \right)$, is defined by [21,22]
(5)$$M^{*}\left(\omega \right)=\;M^{\prime }\left(\omega \right)+i\;M^{\prime \prime }\left(\omega \right)=\frac{1}{\varepsilon^{*}\left(\omega \right)}={i\omega Z}^{*}\left(\omega \right).$$

We established the scientific performance of the LM_x_F_2−x_PO glasses by synthesizing the glass samples and analyzing the ionic conductivities of them. We applied three different formalisms to analyze the performance of the ionic conductivities of the LM_x_F_2−x_PO glasses. The overall research stream is schematically described in Figure 1.

## 3. Results and Discussion

Figure 2 shows the selected XRD patterns of the 3Li_2_O-xMn_2_O_3_-(2-x)Fe_2_O_3_-3P_2_O_5_ (LM_x_F_2−x_PO; x = 0, 0.5, 1.0, 1.6, 2.0) glass samples at room temperature. The XRD patterns were found to show wide halos without crystalline sharp Bragg peaks in the range of 10–80°. The broad XRD patterns indicate the nature of the amorphous structure. The slightly different XRD patterns for LM_x_F_2−x_PO glasses can be understood based on the real space atomic distributions (RDF) [26]. The XRD patterns are momentum transfer of the scattered X-rays, which is the Fourier transformation of RDF. As a change of the mixed iron–manganese oxide content in LM_x_F_2−x_PO glasses, the local variations of any atomic characteristics appear in X-ray scattering. Detailed measurements and calculations of RDF for LM_x_F_2−x_PO glasses are not the schemes of this research, but we can figure out the physical properties that the change in RDF occurs from the depolymerization of the network with the hybrid of iron and manganese in the phosphate glass. As a consequence, the results of the modification of the local structure will respond to the variation of conductivity.

Figure 3 shows the DTA curves for LM_x_F_2−x_PO (x = 0, 0.5, 1.0, 1.6, 2.0) glasses, measured at a heating rate of 10 °C/min. The specific temperatures of T_g_, T_c_, T_m_, and T_x_ were determined from the extrapolation method as shown in the figure. The glass transition accompanies a drastic variation of viscosity, leaving the disordered network structure almost as it is. Meanwhile, the crystallization from a disordered glass state to an ordered crystalline state involves a discontinuous change in order parameters. These transitions accompany the variation of heat capacities which can be measured by DTA experiments.

In various metallic and oxide glasses, the glass-forming ability (GFA) is defined as T_c_/(T_g_ + T_m_), relating to devitrification, melting and amorphization processes. The relationship of time-temperature-transformation for different phases indicated that the lower values of GFA provided a smaller value in GFA [27]. The calculated values of GFA for the LM_x_F_2−x_PO glasses are shown in Table 1.

Hudgens and Martin showed that the structure of amorphous P_2_O_5_ was suggested to consist of a 3-D structure of tetrahedral P=O units and the network was modified by the occurrence of non-bridging oxygens (NBOs) which induced a decrease in T_g_ [28]. As shown in Table 1, the T_g_ (419 °C) of the manganese-free sample, LM_x_F_2−x_PO (x = 0) is higher than the T_g_ (370 °C) of iron-free glass, LM_x_F_2−x_PO (x = 2), implying that the modification of the network by cation-anion atomic linkage in the LM_x_F_2−x_PO (x = 2) is stronger than the modification by those in the LM_x_F_2−x_PO (x = 0). It is observed that the T_g_ of LM_x_F_2−x_PO (0.25≤x≤2.0) glasses decreases from 439 °C to 370 °C with further Mn_2_O_3_ additions. Regarding the decrease of T_g_ for the mixed iron–manganese component in LM_x_F_2−x_PO (0.25≤x≤2.0) glasses, we expect that the structural modification of the mixed transition metallic effect plays a dominant role, because the molar concentration of lithium element is constant in LM_x_F_2−x_PO glasses.

Lai and co-workers reported that the crystal structure of (Mn_x_Fe_1−x_)_2_O_3_ nanoparticles was single-phase compounds with low Mn(III) content but it was changed to a mixture of two separated phases when Mn_2_O_3_ is the majority oxide, i.e., from 78 to 89 Mn(III) atom % [11]. Thus, it is interesting to figure out how the electrical conductivity has changed with Mn content in the glass.

For the manganese-free sample, LM_x_F_2−x_PO (x = 0), authors showed in a previous paper that the depolymerization increased by the structural change with having a P-O-Li^+^ group in the glass network and by the change in bonding forces of iron valence [15]. Similarly, the samples exhibit a lower GFA with increasing content of the Mn_2_O_3_ in LM_x_F_2−x_PO (0.25≤x≤2.0) glasses, appearing that the addition of the Mn_2_O_3_ change the scope of the P-O-P bonds in the glass network. It is known that the ionicity of manganese varies by +2, +3, +4, and +7 for the surrounding environment and the ionicity of iron changes by +2 and +3 in the neighboring environment. As a consequence, the manganese oxide in LM_x_F_2−x_PO 0.25≤x≤2.0) glasses makes it easy for restructuring the surrounding glassy network.

Figure 4a exhibits the compositional dependence of complex impedance spectra with Cole–Cole plot, measured at room temperature for LM_x_F_2−x_PO (0≤x≤2.0) glasses. Figure 4b,c shows the temperature-dependent Cole–Cole plot of LM_x_F_2−x_PO for x = 0 and for x = 1.0 glasses, respectively. The estimated value of exponent *δ* from the fits of Equation (1) varies at 0.73 ± 0.02 for LM_x_F_2−x_PO (x = 0) glass and it decreases slightly as 0.71 ± 0.01 for LM_x_F_2−x_PO (0.25≤x≤2.0) glasses in the temperature range from 30 °C to 240 °C. The results imply that the variation of local potential energy depth of the LM_x_F_2−x_PO glasses slightly increases due to the mixed iron–manganese oxide elements. In addition, the measured *δ* is almost independent of temperature for all samples, suggesting that the thermal energy contributes to speeding up the ionic hops and transportations. The ionic dynamics can be slightly disturbed because of the mixed iron–manganese oxide element that changes the local structure of the network. The dc-conductivity values of the cellulose-based membranes were also characterized from the Cole–Cole plot [29].

The inset of Figure 4a shows an example that an equivalent circuit describes the electrical response of the LM_x_F_2−x_PO (x = 0.75) glass. The corresponding data fit for the electrochemical impedance spectroscopy is R = 89.6 MΩ and C = 2.9 pF for the LM_x_F_2−x_PO (x = 0.75) sample. The Cole–Cole impedance Z* in Equation (1) is equivalent to the expression of RC = (iω)^δ−1^τ^δ^. A distorted semicircle arc in Figure 4a intersects at the real axis of the low-frequency region of the complex impedance, which gives the dc conductivity, $\sigma_{\mathrm{dc}}=\frac{L}{Z_{0}A}=\frac{1}{Z_{0}^{*}}$. The obtained value of the impedance at zero frequency is Z_0_′(x = 0) = 1.15 × 10^7^ S^−1^ for LM_x_F_2−x_PO (x = 0) glass and it increases gradually up to Z_0_′(x = 2.0) = 2.90 × 10^8^ S^−1^ with increasing manganese element in LM_x_F_2−x_PO (0.25≤x≤2.0) glasses, indicating that the dc conductivity decreases gradually with increasing content of the Mn_2_O_3_ in LM_x_F_2−x_PO glasses. Namely, the dc conductivity $\sigma_{\mathrm{dc}}$ = 8.70 × 10^−8^ S/cm for LM_x_F_2−x_PO (x = 0) glass decreases to 3.45 × 10^−9^ S/cm for LM_x_F_2−x_PO (x = 2.0) glass, showing that the dc conductivity ratio between LM_x_F_2−x_PO (x = 0) glass and LM_x_F_2−x_PO (x = 2.0) glass turns out to be $\sigma_{\mathrm{dc}}\left(x=2.0\right)/\sigma_{\mathrm{dc}}\left(x=0\right)$ = 0.04.

Figure 4d shows that the conductivities of dc and ac obey the Arrhenius $\sigma_{\mathrm{dc}}T=C^{\prime }\exp\left(-\frac{E_{\mathrm{dc}}^{C}}{k_{B}T}\right)$ and $\omega_{p}=\omega_{0}\exp\left(-\frac{E_{\mathrm{ac}}^{C}}{k_{B}T}\right)$, where $E_{\mathrm{dc}}^{C}$ and $E_{\mathrm{ac}}^{C}$ are the dc and ac activation energies, respectively. $C^{\prime }$ is a constant, $\omega_{0}$ is the onset frequency and k_B_ is the Boltzmann constant. The peak frequency $\omega_{p}$ satisfies $\omega_{p}\tau =1$ [23] and $\omega_{p}$ is defined as the maximum value of the imaginary part of the complex impedance plot. Figure 4d shows the overall decrease in peak frequency values with the increase of *x* in the LM_x_F_2−x_PO glasses, which represents that the atomic or molecular relaxation times are gradually longer, with an increasing amount of manganese oxide.

The slopes of the conductivity against temperature represent the values of the activation energies. Dc activation energy represents that the charge carriers are moving through a long distance over the heights of the random energy barriers, and the ac activation energy represents that the charge carriers are moving back and forth through the potential barriers of short-range. As shown in Table 2, the complex impedance activation energies of $E_{\mathrm{dc}}^{C}$ and $E_{\mathrm{ac}}^{C}$ increase gradually with increasing manganese oxide components in the LM_x_F_2−x_PO glasses, indicating that both conductivity of dc and ac decrease with increasing Mn_2_O_3_ component due to the mixed potential barriers of asymmetric depth.

In summary, we observe the properties of the LM_x_F_2−x_PO glasses such that the value of *δ* decreases but both activation energies of dc and ac increase with increasing the manganese component in the LM_x_F_2−x_PO glasses. We may interpret the complex impedance observations in terms of the free energy landscape. The ion encounters the asymmetric potential sites that are irregular, resulting from a more broaden distribution in barrier heights and a different barrier depth caused by the mixed iron–manganese oxide component in the LM_x_F_2−x_PO glasses. Consequently, more energy of migration for ionic charge carriers may be needed to hop through the asymmetrically distorted potential barriers, leading to low ionic conductivity.

Figure 5a shows the conductivity spectra of the real value, $\sigma \left(\omega \right)$, which was measured at room temperature for the LM_x_F_2−x_PO glasses. We obtain that the dc conductivity $\sigma_{\mathrm{dc}}$ = 8.70 × 10^−8^ S/cm for LM_x_F_2−x_PO (x = 0) glass increases as to 3.50 × 10^−9^ S/cm for LM_x_F_2−x_PO (x = 2.0) glass, exhibiting that $\sigma_{\mathrm{dc}}\left(x=2.0\right)/\sigma_{\mathrm{dc}}\left(x=0\right)$ = 0.04. It is shown that the result obtained from the power–law analysis is very similar to the one obtained from the Cole–Cole representation. A temperature-dependent Arrhenius plot is shown in Figure 5b, $\sigma_{\mathrm{dc}}T=C^{\prime \prime }\exp\left(-\frac{E_{\mathrm{dc}}^{J}}{k_{B}T}\right)$, which is obtained from the fit of the power–law in Equation (2).

The obtained values of $E_{\mathrm{ac}}^{J}$ from the temperature-frequency relationship of the experimental data to Equation (3) are shown in Table 2. It is noted that the estimated activation energies, from the Cole–Cole function and the universal power–law equation, are very close to each other. The result indicates that the conduction mechanism is consistent with the electrical relaxation.

The average value of power–law exponent *s* in Equation (2) is 0.59 ± 0.01 for the LM_x_F_2−x_PO (x = 0) glass and 0.60 ± 0.01 for the LM_x_F_2−x_PO (0.25≤x≤2.0) glasses, suggesting that the value of *s* is almost independent of composition, as well as the temperature of the samples. The result represents that the ions are moving through the conduction pathway with fractal dimension D_f_ = 1.36 for *s* = 0.6. The corresponding conduction pathway is independent of the composition and temperature, since the exponent, *s*, is expressed as *s =* 4/3 − 1/D_f_ from the derivation of power–law based on the modified fractional Rayleigh equation [24].

In the following, we will show how the hopping conductivity of ions applies to the mixed iron–manganese–phosphate oxide network system. For example, we may estimate the number of mobile Li-ions, participating in the ionic hopping through the LM_x_F_2−x_PO glasses, i.e., ρ_m_(80 °C) ≅ 3.40 × 10^16^ in the LM_x_F_2−x_PO (x = 2.0) glass. For calculation, we use the atomic mass M = 6.94 g/mol for Li, M = 16.0 g/mol for O, M = 54.94 g/mol for Mn, M = 55.85 g/mol for Fe, M = 30.97 g/mol for P, and the Avogadro’s number N_A_ = 6.02 × 10^23^ per mole, the measured exponent s = 0.60, and the hopping frequency $\omega_{h}$ = 1.34 × 10^5^ rad/s at T = 80 °C.

Figure 5c,d shows the measured values of dc conductivity,$\;\sigma_{dc}$, and the calculated values of cation number density, $\rho_{m}$, as a function of the mixed concentration of an iron–manganese x, at various temperatures. The slope of ln$\sigma_{dc}$ and ln$\rho_{m}$ decreases systematically as a function of mixed content of x at various temperatures, and the ratio of slope varying between the dc conductivity and the cation number density at a given temperature, ${\ln\sigma }_{\mathrm{dc}}\left(x_{T}\right)/{\ln\rho }_{m}\left(x_{T}\right)$, decreases from 0.97 ± 0.04 to 0.60 ± 0.03 as an increase in the temperature, ranging from the ambient atmosphere to 160 ℃, but the ratio of slope increases to the value of 1.06 ± 0.04 at 160 °C and 0.80 ± 0.04 at 240 °C, respectively. The result indicates that the electrical conductivity basically originates from the number of hopping cations. The cation number density, ρ_m_, in Equation (4) is mostly connected to the representation of the ac conductivity of the system, but the results show that the effect of decreased ρ_m_ is also concerned with decreasing the dc conductivity.

In a previous paper for the lithium-iron-phosphate glasses, we referred that the migration of Li ions was accelerated through the way of cooperative jumps in the thermal fluctuating NBOs sites. This phenomenon is a result of high kinetic energy to make the hops fast between NBOs sites [15]. Similar to the case of the lithium-iron-phosphate glasses, the linearity of conductivity with cation concentration is characteristic of strong forward correlation where some parts of the lithium cations are thought to be participated to jump through NBOs sites in the LM_x_F_2−x_PO glasses.

Furthermore, Cramer and co-authors reported that the ionic conduction in the phosphate-vanadate glass was characterized by the unified site relaxation, in which the site of a mobile ion consisted of a combination of the fast and slow dynamics [30]. They presented conductivity spectra of the phosphate-vanadate glass, covering the very high frequency range of more than 13 decades. Unlikely the phosphate-vanadate glass, the ion dynamics in the LM_x_F_2−x_PO glasses, covering a frequency range of seven decades, do show only the behavior of power–law.

Figure 5 shows the compositional variation of GFA, dc- and ac-activation energies for all glass compositions at a particular temperature in the LM_x_F_2−x_PO glasses. In the thermal analysis section, although Hudgens and Martin argued that the 3D network of P_2_O_5_ was modified by the formation of non-bridging oxygens (NBOs), which caused a large decrease in T_g_, however, it was found that the phosphate glass systems T_g_ cannot reflect the GFA effectively [27].

As shown in Figure 6, the curve of GFA follows a similar fashion of dc- and ac-activation energies, $E_{\mathrm{dc}}^{J}$ and${\;E}_{\mathrm{ac}}^{J}$, of the power–law conductivity. They are constant with the Mn_2_O_3_ content of 0≤x≤0.5 and increase gradually with increasing manganese oxide component of 0.5≤x≤0.75, then they are constant again with Mn_2_O_3_ content of 0.75≤x≤2.0 in the LM_x_F_2−x_PO glasses. Since the activation energy in the LM_x_F_2−x_PO glasses stands for the behavior of ionic motions in the potential well. Therefore, a new concept of GFA = T_c_/(T_g_ + T_m_) establishes the nature of the structural modification by the NBOs in the LM_x_F_2−x_PO glasses. As an increase in the manganese content of LM_x_F_2−x_PO glasses, the local variation of the atomic distance in the structural units of LM_x_F_2−x_PO glasses is expected to be very complicated because the change of ionicity in the manganese-iron compound can modify the glass network more diverse. As shown in Table 1, the value of GFA is not linearly dependent on Mn_2_O_3_ concentration from x = 0.5 to x = 0.75. It was shown that the crystal structure of mixed iron–manganese oxide nanoparticles changed with Mn concentrations from 12 to 44 atom %, as well as from 78 to 89 atom % [11]. That is, the crystal phase separation depends on the Mn concentration of mixed iron–manganese oxide nanoparticles. These properties are very similar to the characteristics of dc and ac activation energies, and GFA in LM_x_F_2−x_PO glasses. Correspondingly, the curves of activation energies and GFA of the glasses show a non-linear dependency in a certain range of concentration of Mn_2_O_3_, reflecting the complexity of the network structure in the LM_x_F_2−x_PO glasses.

The master curves of the conductivity are shown in Figure 7 for all LM_x_F_2−x_PO glasses at an ambient temperature, which curves are superimposed onto a single master curve. The scaling conductivity isotherm in such a plot along a line of the slope with the onset hopping frequency of conductivity dispersion, σ(ω_h_) = 2σ_dc_. The shape of σ(ω) overlaps very well below T_g_, implying that the transport mechanism is operating in a common pathway regardless of the mixed compositions of iron oxide and manganese oxide. It is shown in the section on power–law analysis that the ions are moving through the conduction pathway with fractal dimension D_f_ = 1.36 for all samples of LM_x_F_2−x_PO glasses.

Figure 8 shows the master plots for the electrical modulus, $\;M^{\prime \prime }\left(\omega \right)/M^{\prime \prime }{\left(\omega \right)}_{\max}$ vs. $\log\left(\frac{\omega }{\omega_{\max}}\right)$ for the LM_x_F_2−x_PO (0≤x≤2.0) glasses at 60 °C and at 120 °C (inset). The low frequency region below $M^{\prime \prime }{\left(\omega \right)}_{\max}$ is attributed with dc conductivity and the high frequency region above $M^{\prime \prime }{\left(\omega \right)}_{\max}$ is associated with the site relaxation inside of potential barriers [31,32,33].

The full width at half-maximum (FWHM) of the modulus representation for the LM_x_F_2−x_PO glasses exhibits the substantial broadening for the samples with the addition of Mn_2_O_3_, because the interaction changes in the mixed iron–manganese–phosphate network due to the variety of the ionicity of manganese of the surrounding glassy network [34]. Correspondingly, the relaxation process and ac conductivity show a big difference in increasing the Mn_2_O_3_ component in the LM_x_F_2−x_PO glasses due to the motion of swamped ions in the asymmetrically distorted potential barriers.

## 4. Conclusions

We studied the effect of compositional changes observed by the addition of Mn_2_O_3_ to the Li_2_O-Fe_2_O_3_-P_2_O_5_ glass on the electrical conductivity and dielectric properties. The observed decrease in dc conductivity and increase in activation energy with the addition of Mn_2_O_3_ is attributed to the formation of ion conducting channels arising from the structural modification and formation of the P-O-Li^+^ linkages, resulting in easy mobility of Li^+^ ions along these bonds. The easy migration of Li^+^ ions through the conduction pathway of the NBO sites that are created by the depolymerization of the mixed iron–manganese–phosphate network and the change of the Mn-Fe-O bond valence in iron–manganese–phosphate clusters. At higher Mn_2_O_3_ content glass, the potential barrier distribution becomes more broadened by different depths of the potential well, and ionic conductivity is hindered as a consequence of the swamped ions in the potential well. The ionicity of manganese can make it easy for restructuring the surrounding glassy network.

The rate of decrease in the estimated cation number density of hopping ions is much more rapid compared with the rate of decrease in electrical conductivity, reflecting that many Li-ions are trapping in the deep asymmetric potential well. The understanding of ionic transport dynamics is a promising effort to develop the functional materials of electrochemical energy storage.

## Figures and Tables

**Figure 1 materials-15-08176-f001:**
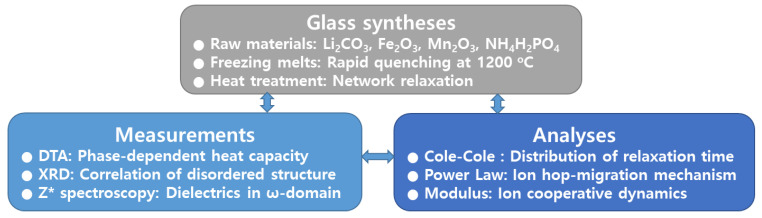
Schematic picture of experimental processes of 3Li_2_O-xMn_2_O_3_-(2-x)Fe_2_O_3_-3P_2_O_5_ glasses.

**Figure 2 materials-15-08176-f002:**
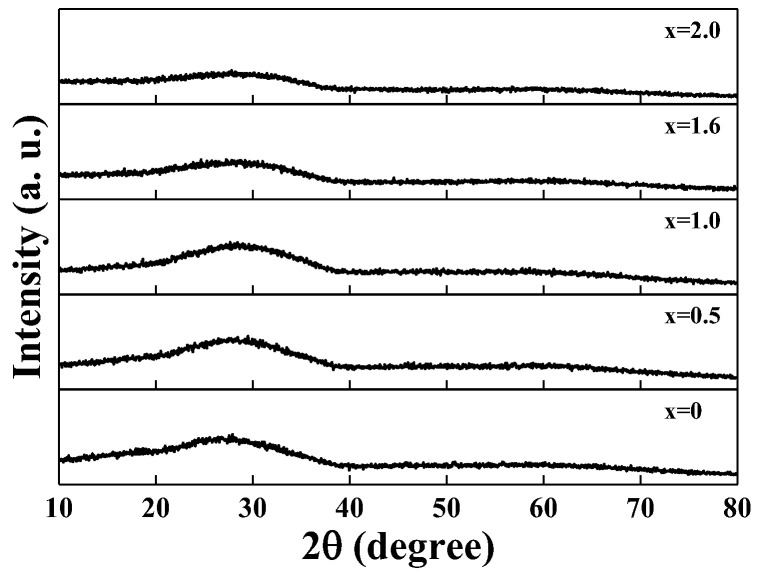
Selected XRD patterns of 3Li_2_O-(2-x)Fe_2_O_3_-xMn_2_O_3_-3P_2_O_5_ (x = 0, 0.5, 1.0, 1.6, 2.0) glasses taken at ambient temperature.

**Figure 3 materials-15-08176-f003:**
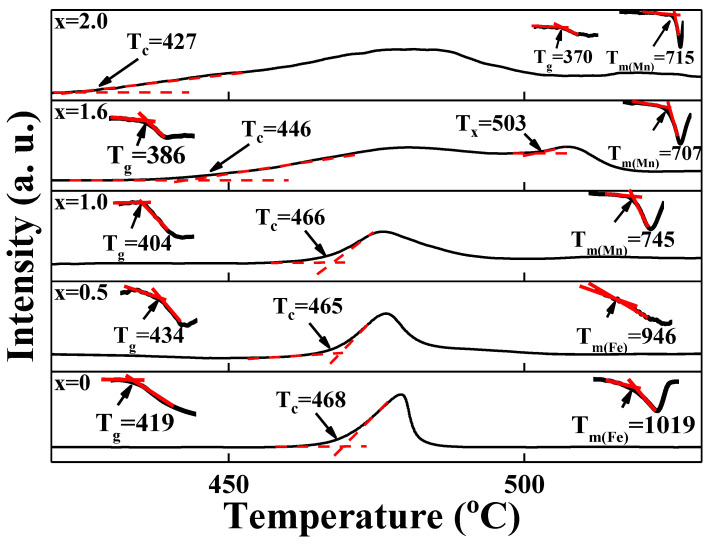
Selected DTA curves for 3Li_2_O-(2-x)Fe_2_O_3_–xMn_2_O_3_-3P_2_O_5_ (x = 0, 0.5, 1.0, 1.6, 2.0) glasses, measured at a heating rate of 10 °C/min. The glass transition temperature is denoted as T_g_, the onset crystallization temperature as T_c_, and the onset melting temperature as T_m_. T_x_ is related to the relaxation temperature of crystallization and/or recrystallization.

**Figure 4 materials-15-08176-f004:**
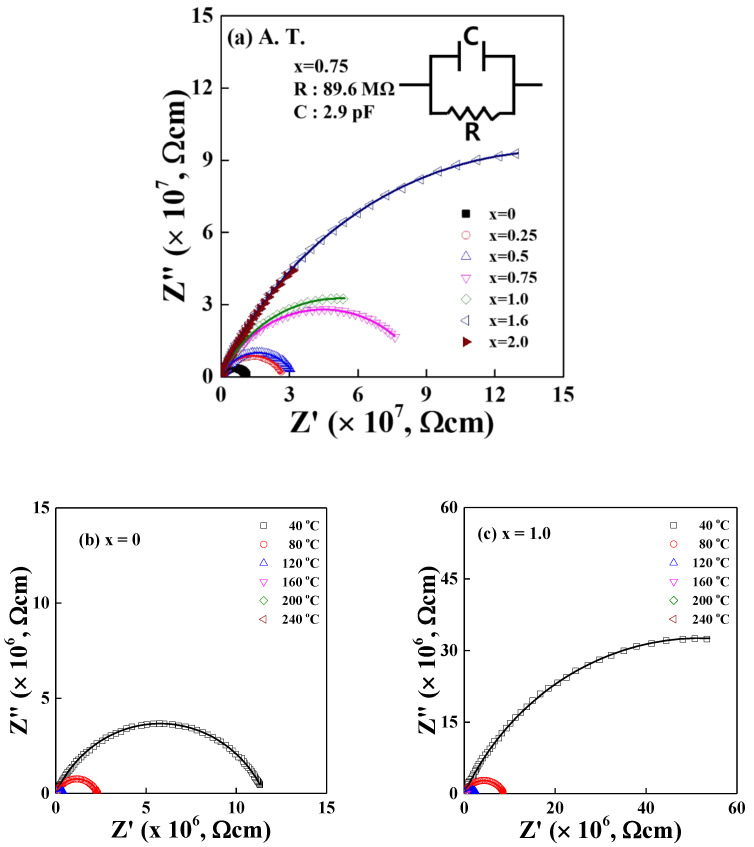
(**a**) Cole–Cole impedance plot of 3Li_2_O-(2-x)Fe_2_O_3_-xMn_2_O_3_-3P_2_O_5_ (x = 0, 0.25, 0.5, 0.75, 1.0, 1.6, 2.0) glasses, measured at ambient temperature. The inset shows an equivalent circuit that describes the electrical response of the LM_x_F_2−x_PO (x = 0.75) glass. (**b**) Temperature-dependent Cole–Cole plot of LM_x_F_2−x_PO (x = 0) glass. (**c**) Temperature-dependent Cole–Cole plot of LM_x_F_2−x_PO (x = 1.0) glass. The intersection of the low frequency side of the complex impedance plot with the *Z’* axis gives the *Z_0_*, which is the impedance of bulk at zero frequency. (**d**) Plots of ln σ_dc_T vs. 1000/T and ln ω_p_ vs. 1000/T. The dc conductivity is taken by 1/Z_0_ and the ac relaxation peak frequency ω_p_ is obtained from fit to the imaginary complex impedance Z” (ω). The slope of solid lines represents dc and ac electrical activation energies, denoted as $E_{\mathrm{dc}}^{C}$ and $E_{\mathrm{ac}}^{C}$, respectively.

**Figure 5 materials-15-08176-f005:**
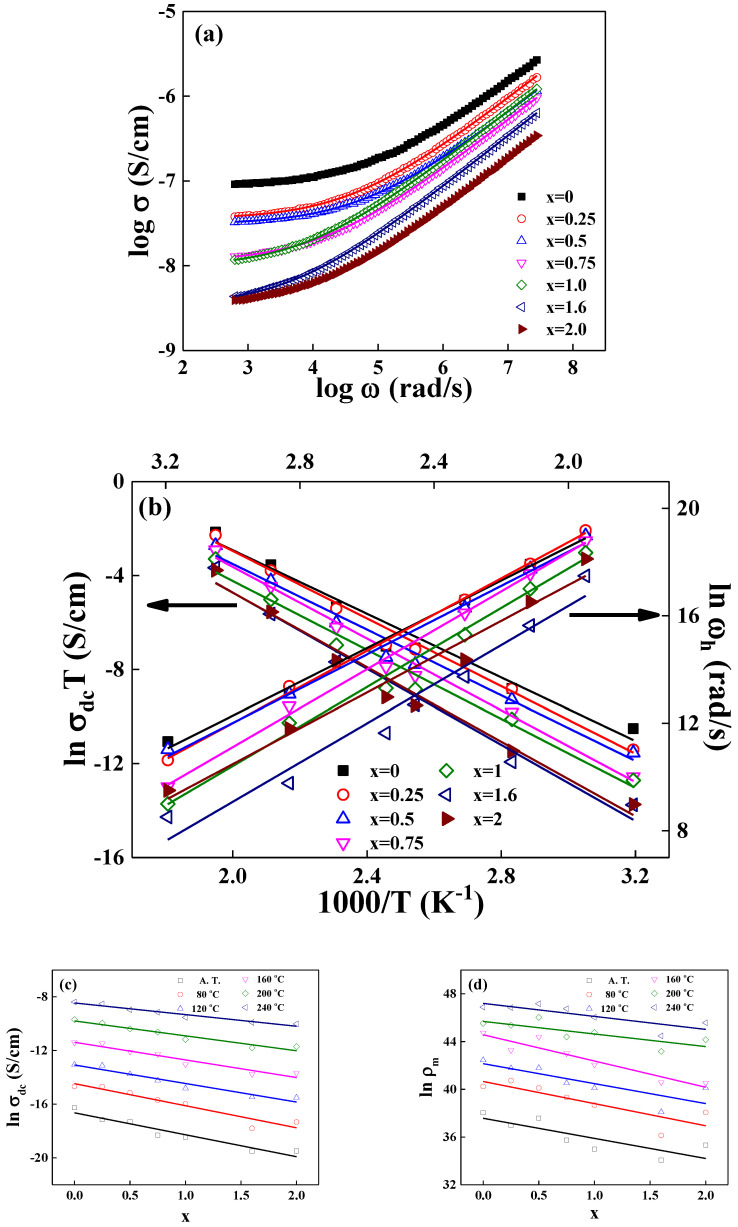
(**a**) Power–law frequency spectra of the real conductivity $\sigma \left(\omega \right)$ for 3Li_2_O-(2-x)Fe_2_O_3_-xMn_2_O_3_-3P_2_O_5_ (x = 0, 0.25, 0.5, 0.75, 1.0, 1.6, 2.0) glasses, measured at ambient temperature. (**b**) Plots of ln σ_dc_T vs. 1000/T and ln ω_h_ vs. 1000/T. The hopping frequency $\omega_{h}$, obeying the Arrhenius relation $\omega_{h}=\omega_{0}\exp\left(-\frac{E_{\mathrm{ac}}^{J}}{k_{B}T}\right)$, is obtained from a fit to $\sigma \left(\omega_{h}\right)=2\sigma_{dc}$. (**c**) Plot of ln$\sigma_{\mathrm{dc}}$ vs. x at various temperatures. (**d**) Plot of ln$\rho_{m}$ vs. x at various temperatures. The solid lines are the least squares straight line fits to the data.

**Figure 6 materials-15-08176-f006:**
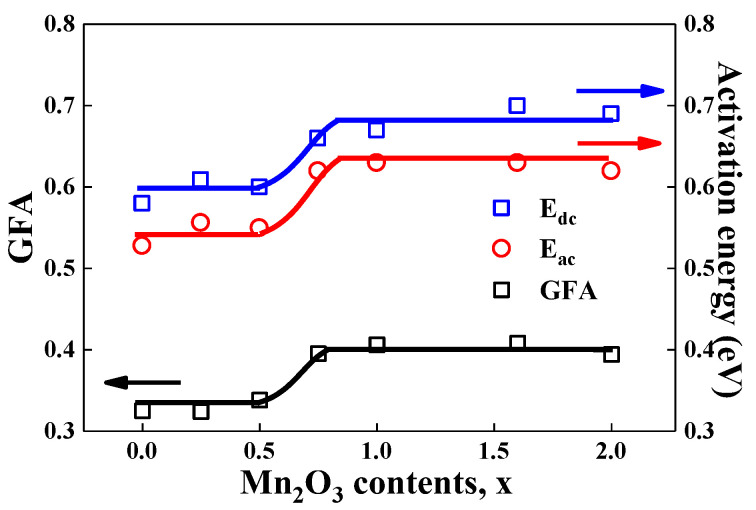
Composition dependence of GFA = T_c_/(T_g_ + T_m_) and dc- and ac-activation energies,$\;E_{\mathrm{ac}}^{J}$ and $E_{\mathrm{ac}}^{J}$, for 3Li_2_O-(2-x)Fe_2_O_3_–xMn_2_O_3_-3P_2_O_5_ (0≤x≤2.0) glasses. The curves are guides for the eyes.

**Figure 7 materials-15-08176-f007:**
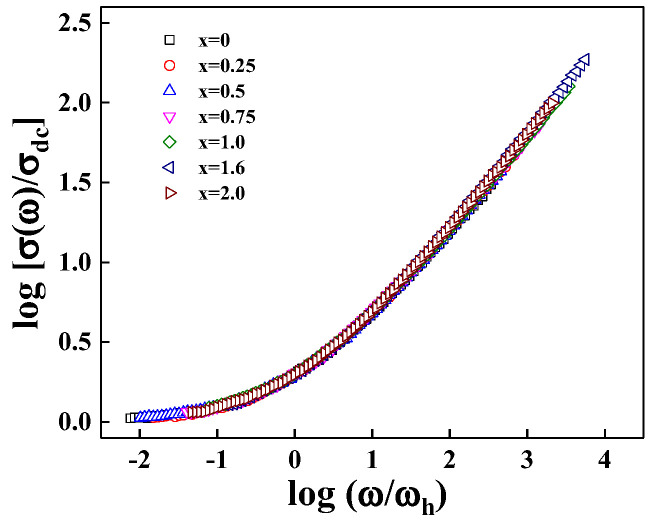
The master curve of 3Li_2_O-(2-x)Fe_2_O_3_-xMn_2_O_3_-3P_2_O_5_ (x = 0, 0.25, 0.5, 0.75, 1.0, 1.6, 2.0) glasses exhibits the fact that the substantially differing ion concentration plot satisfies the common scaling, σ(ω_h_) = 2σ_dc_.

**Figure 8 materials-15-08176-f008:**
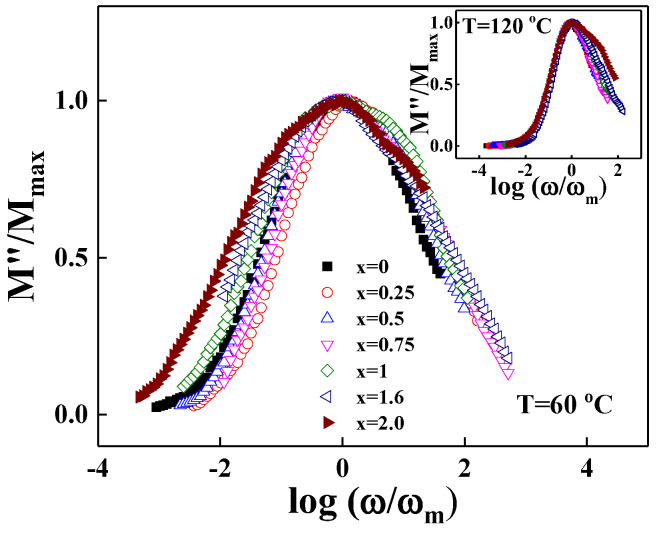
Scaling plot of M”/M”_max_ vs. log(ω/ω_m_) for 3Li_2_O-(2-x)Fe_2_O_3_-xMn_2_O_3_-3P_2_O_5_ (x = 0, 0.25, 0.5, 0.75, 1.0, 1.6, 2.0) glasses at 60 °C. Inset corresponds to the scaling plot at 120 °C.

**Table 1 materials-15-08176-t001:** Composition and thermal properties for the 3Li_2_O-xMn_2_O_3_-(2-x)Fe_2_O_3_-3P_2_O_5_ (x = 0, 0.25, 0.5, 0.75, 1.0, 1.6, 2.0) glasses. Glass-forming ability, GFA = T_c_/(T_g_ + T_m_).

Glass Composition	T_g_ (°C)	T_c_ (°C)	T_m_ (°C)	GFA
x = 0	419	467	1019 (Fe)	0.325
x = 0.25	439	460	980 (Fe)	0.324
x = 0.5	434	465	946 (Fe)	0.337
x = 0.75	418	459	744 (Mn)	0.395
x = 1.0	404	466	745 (Mn)	0.406
x = 1.6	386	446	707 (Mn)	0.408
x = 2.0	370	427	715 (Mn)	0.394

**Table 2 materials-15-08176-t002:** Composition and thermal dependent activation energies for the 3Li_2_O-xMn_2_O_3_-(2-x)Fe_2_O_3_-3P_2_O_5_ (x = 0, 0.5, 0.75, 1.0, 2.0) glasses. Activation energies E^C^ for Cole–Cole and *E^J^* for Jonscher representation, respectively.

Glass Composition	$\sigma_{dc}^{C}$ $(S/m),\;\sigma_{dc}^{J}$ $(S/m);\;\omega_{p}$ $\left(\mathrm{Hz}),\;\omega_{h}(\mathrm{Hz}\right)$			E^C^_dc_ (eV)	E^C^_ac_ (eV)
	A.T.	120 °C	240 °C	E^J^_dc_ (eV)	E^J^_ac_ (eV)
x = 0	8.7 × 10^−8^, 8.7 × 10^−8^; 4.7 × 10^4^, 8.2 × 10^4^	2.1 × 10^−6^, 2.1 × 10^−6^; 4.8 × 10^6^, 2.9 × 10^6^	2.3 × 10^−4^, 2.3 × 10^−4^; 6.5 × 10^7^, 1.9 × 10^8^	0.58 ± 0.01 0.58 ± 0.01	0.51 ± 0.02; 0.54 ± 0.01
x = 0.5	3.1 × 10^−8^, 3.1 × 10^−8^; 1.3 × 10^4^, 6.1 × 10^4^	1.1 × 10^−6^, 1.1 × 10^−6^; 3.5 × 10^5^, 1.9 × 10^6^	1.3 × 10^−4^, 1.3 × 10^−4^; 4.4 × 10^7^, 1.7 × 10^8^	0.60 ± 0.01 0.60 ± 0.01	0.55 ± 0.01 0.55 ± 0.01
x = 0.75	1.1 × 10^−8^, 1.1 × 10^−8^; 3.8 × 10^3^, 1.5 × 10^4^	6.6 × 10^−7^, 6.6 × 10^−7^; 2.2 × 10^5^, 1.3 × 10^6^	1.1 × 10^−4^, 1.1 × 10^−4^; 3.4 × 10^7^, 1.5 × 10^8^	0.66 ± 0.01 0.66 ± 0.01	0.61 ± 0.01 0.62 ± 0.01
x = 1.0	9.6 × 10^−9^, 9.6 × 10^−9^; 6.8 × 10^2^, 8.1 × 10^4^	3.7 × 10^−7^, 3.7 × 10^−7^; 1.2 × 10^5^, 6.0 × 10^5^	7.2 × 10^−5^, 7.2 × 10^−5^; 2.5 × 10^7^, 9.3 × 10^7^	0.67 ± 0.03 0.67 ± 0.03	0.67 ± 0.02 0.63 ± 0.02
x = 2.0	3.4 × 10^−9^, 3.5 × 10^−9^; 7.0 × 10, 1.3 × 10^4^	1.8 × 10^−7^, 1.8 × 10^−7^; 5.4 × 10^4^, 4.3 × 10^5^	4.5 × 10^−5^, 4.5 × 10^−5^; 1.3 × 10^7^, 7.4 × 10^7^	0.69 ± 0.03 0.69 ± 0.03	0.69 ± 0.02 0.62 ± 0.02

## Data Availability

Not applicable.

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
