# Peer review of "Characterization of Ionic Transport in Li2O-(Mn:Fe)2O3-P2O5 Glasses for Li Batteries"

_materials, 2022, doi:10.3390/ma15228176_

Round 1

Reviewer 1 Report

In this work, the authors present a systematic study of the lithium-ion transport upon the mixed manganese-iron oxide phosphate glasses 3Li2O-xMn2O3-(2-x)Fe2O3-3P2O5(LMxF2-xPO ;0≤ ? ≤2.0) proposed for the use in a cathode for lithium-ion batteries. The process of the ionic conduction and the relaxation in LMxF2-xPO glasses are analyzed. Based on the journal scope and high quality, some revisions should be made before this manuscript can be accepted after addressing the following questions.

1. The authors and the previous reports agreed that the content of Mn ion has a great influence on the electrical conductivity due to the different Mn content in the glass. The author must comment on the real contents of Mn (or Mn2O3) in the LMxF2-xPO glasses samples. For example, can the author measure and analysis it by ICP or other quantitative detection method?

2. In Figure 3A,the equivalent electrical model that adopted for EIS data fit and interpretation is need.

3. The XRD image in Figure 1 should be augmented with the XRD standard card.

4. The graph should be legible. For example, some text is not legible in the Figure 2; The lines overlap without distinguishing effectively for Figure 3(b) and Figure 4(b), etc.

5. Figure 7 only one gragh, so it be need to remove the (a).

Reviewer 2 Report

This paper experimentally studied the effect of compositional changes on the electrical conductivity and dielectric properties. The result is interesting and helpful to improve the battery performance. However, there have some minor issues should be addressed before publication. (1) Since there have some literatures conducted the similar work, the novelty should be highlighted and the question to be solved should be described in the introduction. (2) Figure and Table are suggested in section 2 to clearly understand the experiment process and the equipment used in this paper. (3) Reviewer the performance indicators, as well as the relative equations, should be introduced in section 2 to make the structure of paper more clear. (4) In results section, more deep analysis should be added to explain the reason of the curve, rather than only describing the phenomenon. (5) The language should be improved.

Reviewer 3 Report

The electrical conductivity and dielectric characteristics of mixed manganese-iron oxide phosphate glasses are reported in this publication. This study appears to be of importance to the material science community, particularly in terms of academic standpoint. The paper may be published in Materials, however, after some important corrections, as justified in the following points:

1. Figure 3a: for a better visualization of the experimental points and related fits, the Cole-Cole plots should be orthonormal. Also, Cole-Cole plots selected at various temperatures for a representative sample should be presented. A related work should be referenced to enrich this part: Applied Surface Science, 607, 155077 (2023).

2. Figure 3b: The numerical values of dc conductivity are not clearly seen. The linear fittings appear to cover them. Please highlight them.

3. Please make the connection between the literature and numerical values of dc and ac electrical activation energy.

4. The authors noted on page 14 that the activation energy values calculated from the Cole-Cole representation and the universal power-law formalism are similar. Based on this discovery, they concluded that the activation energy is an adequate parameter for defining the electrical relaxation and conduction pathways. How did they arrive at the latter conclusion? Please elaborate.

5. On page 20, sentence: “The full width at half-maximum…”. It is a too complicated sentence. Please revise it and check the grammar. A reference is required here.

Round 2

Reviewer 1 Report

No problem with publishing.

Reviewer 3 Report

I agree with the revision version of the manuscript. In my opinion, it may be accepted in present form.

A small observation: the reference 29 is added for two times at the "Reference" section. Please remove one of them.